# CatCode: A Comprehensive Evaluation Framework for LLMs On the Mixture of Code and Text

**Zhenru Lin**
IIIS, Tsinghua University
lzr22@mails.tsinghua.edu.cn

**Yiqun Yao**
Beijing Academy of Artificial Intelligence
yqyao@baai.ac.cn

**Yang Yuan***
IIIS, Tsinghua University
Shanghai Artificial Intelligence Laboratory
Shanghai Qi Zhi Institute
yuanyang@tsinghua.edu.cn

## Abstract

Large language models (LLMs) are increasingly proficient in understanding and generating a mixture of code and text. Evaluation based on such *mixture* can lead to a more comprehensive understanding of the models' abilities in solving coding problems. However, current evaluation methods are either limited in task coverage or lack standardization. To address this issue, we propose to apply category theory as math abstraction for code-related evaluation. Specifically, morphisms within a code category can represent code debugging and transformation, functors between two categories represent code translation, and functors between a code category and a natural language category represent code generation and explanation. We present an automatic evaluation framework called **CatCode** (**Cat**egory **Code**) that can assess the coding abilities of various ChatGPT-like LLMs in a *comprehensive* and *standard* way, and further support *composite* task evaluation. The code can be found at https://github.com/scorpio-nova/CatCode.

## 1 Introduction

The success of large language models (LLMs) as programming assistants has been widely acknowledged, with their higher proficiency demonstrated in various coding tasks such as code generation (Li et al., 2022; Svyatkovskiy et al., 2020), code explanation (MacNeil et al., 2022), and code translation (Zhu et al., 2022). However, what truly sets the recent LLMs apart is their ability, akin to ChatGPT-like models, to comprehend and align with human intents by processing a *mixture* of natural language and code. This unique attribute significantly lowers the entry barrier for users, leading to their widespread adoption and notable achievements.

To assess the coding ability of LLMs, numerous efforts have been made by researchers. However, current evaluation methods are either limited in task coverage or lack standardization. Match-based automatic frameworks such as CodeBLEU (Ren et al., 2020) rely primarily on similarity scores to evaluate the quality of code. However, such frameworks often fail to capture the nuances of code functionality and meaning. Execution-based evaluation methods, for example, MBXP (Athiwaratkun et al., 2022) and MultiPL-E (Cassano et al., 2023), can evaluate the function accuracy of code, but they primarily focus on code generation and translation tasks. Task-based evaluation frameworks like CodeXGLUE (Lu et al., 2021) offer a comprehensive approach but lack standardization due to completely different datasets, task formulations, and APIs. Consequently, we still lack an evaluation framework to adapt to the context of a mixture of natural language and code, and there is a need to

---

*Corresponding author.

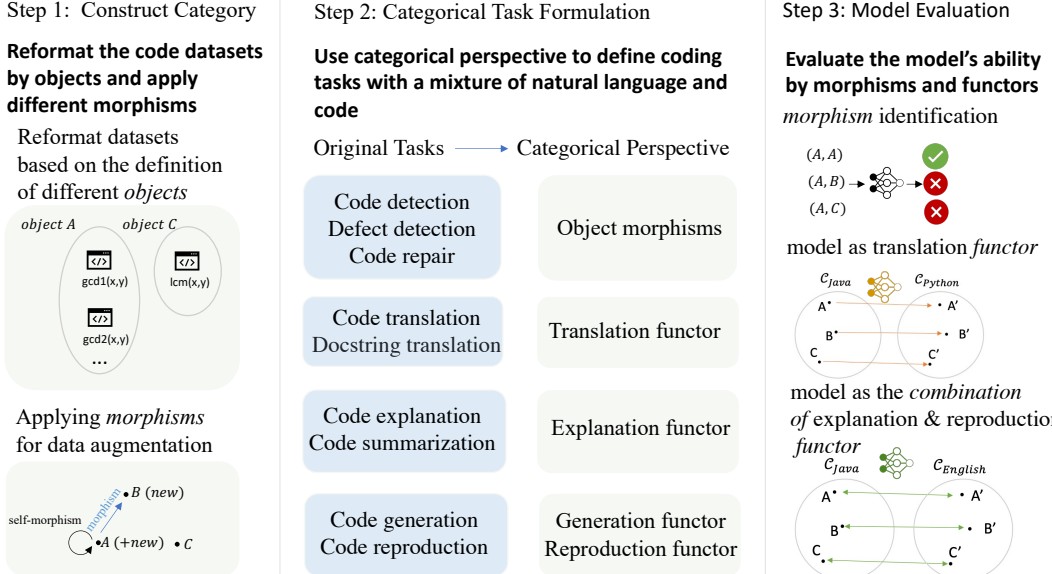

Figure 1: The overall evaluation framework. We use category perspectives to reorganize and transform data, formulate different coding tasks, and conduct model evaluations.

establish a comprehensive evaluation framework that not only supports diverse and novel task definitions but also provides a standardized approach.

We aim to establish a comprehensive theoretical framework that can be open-sourced and applied to essential coding tasks, providing standardized automatic evaluation metrics. It is a challenging task without theoretical guidance. Therefore, we seek a theory that can effectively express the structural aspects of code, language, and their interrelationships. In this context, we find category theory, a branch of mathematics, to be particularly suitable for describing relationships between various elements.

By utilizing category theory's mathematical abstractions, we can gain insights into the relationships among different programming languages (PLs) and natural languages (NLs). We consider PLs and NLs as categories, and functionally equivalent programs as objects, and leverage functors and morphisms to capture the object relations within and across these categories. This gives a unified framework for describing functional equivalence, which not only works within a single PL, but also among different PLs, and even between PLs and NLs. By learning the morphisms within a programming language category, the model can grasp the similarities and differences between code snippets. Additionally, by acquiring knowledge of the functors between categories, the model can understand the relationship between different programming languages and natural languages.

Based on the categorical perspective above, we build a standardized evaluation framework (see Figure 1). It is standardized in terms of data definition, task formulation, and APIs. This framework can be extended to many code-related tasks as long as a categorical definition is given. In our experiments, we give some examples of common code-related tasks, and assess models such as ChatGPT and CodeGeeX for their capabilities in identifying functional equivalence in code, performing code translation, generating code explanations, and reconstructing code based on explanations. We have observed that these models still struggle to differentiate between the concepts of "functional equivalence" and "similarity" in code. While they demonstrated relatively satisfactory performance in code translation, maintaining functional equivalence between code and its corresponding explanations remains a persistent challenge.

Our main contributions can be listed as follows:

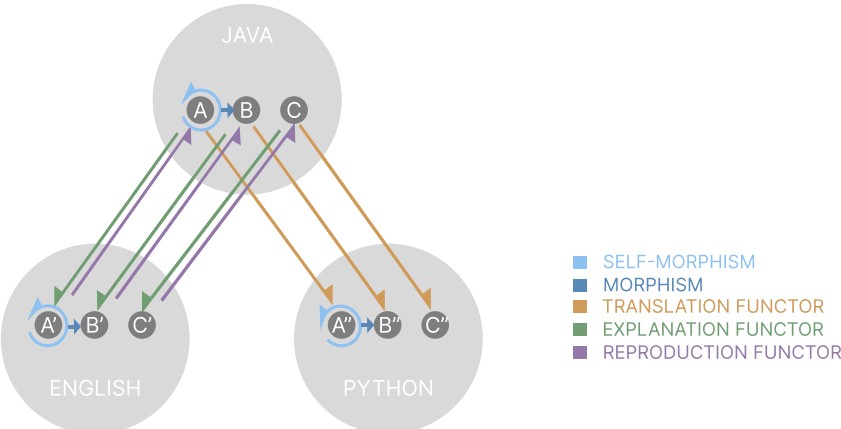

Figure 2: Categorical framework for a mixture of code and NL. *A*, *B* and *C* represent different objects, *A* and *A''* represent the equivalent object of *A* in other categories.

- We introduce **CatCode**, a novel evaluation perspective for code-related tasks based on category theory, which provides a *comprehensive* framework that encompasses a wide range of code-related task formulations.

- We present a *standardized* automatic evaluation platform within the CatCode framework, that offers a quantitative assessment of the coding abilities of Language Models (LLMs) and can adapt to various datasets and models, which will be publicly available.

- We evaluate competitive LLMs, including ChatGPT, StarCoder and CodeGeeX, providing insights into their strengths and limitations in understanding the *mixture* of NL and code.

## 2 Methods

Generally, it is difficult to achieve both comprehensiveness and standardization. In Section 2.1, we show that the categorical perspective, with its emphasis on generalization and abstraction, offers a valuable approach to achieving comprehensiveness in dealing with the mixture of code and natural language. In Section 2.2, we discuss the significance of standardization and outline strategies for achieving it.

### 2.1 Comprehensive Categorical Perspective

Category theory studies how structures arising in different areas can be described with a common language. It was originally developed to transport ideas from one branch of mathematics to another, e.g. from topology to algebra. Category Theory has been applied to different fields, ranging from functional programming to logic, software design, and linguistics(Spivak, 2014; Ahrens & Wullaert, 2022; Fong & Spivak, 2018; Bradley, 2018). Here we provide the basic definitions used in our work and explain their applications to describe code and natural language. Figure 2 illustrates those concepts in a *comprehensive* way.

**Definition 2.1.** (Category, Object, Morphism). A category $\mathcal{C}$ consists of a set of objects $\text{Ob}(\mathcal{C})$ and a set of morphisms $\text{Hom}_{\mathcal{C}}(a, b)$, from $a$ to $b$ for every $a, b \in \text{Ob}(\mathcal{C})$. Given two morphisms $f \in \text{Hom}_{\mathcal{C}}(a, b), g \in \text{Hom}_{\mathcal{C}}(b, c)$, we define their composition as $g \circ f \in \text{Hom}_{\mathcal{C}}(a, c)$. The composition of morphisms is governed by two axioms:

- *Associativity*: If $f : a \to b, g : b \to c$, and $h : c \to d$ then $h \circ (g \circ f) = (h \circ g) \circ f$.

- *Identity*: For every object $x$, there exists a morphism $\text{id}_x : x \to x$ called the identity morphism for $x$, such that for every morphism $f : a \to b$, we have $\text{id}_b \circ f = f = f \circ \text{id}_a$.

**Application 2.1.** (Category, Object, Morphism of Code and NL). For describing the mixture of code and natural languages, we first regard them as objects from different categories. We consider the code categories that contain all executable code in a certain language:

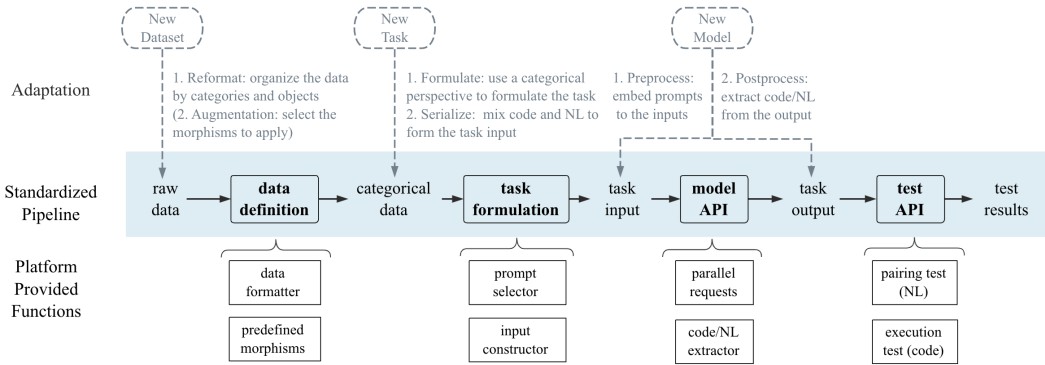

Figure 3: Standardized evaluation platform. The central pipeline offers a consistent approach for all evaluations. Behind the pipeline, we provide a variety of functions to automatically conduct the most important steps. With our platform released, the pipeline can easily accommodate novel datasets, tasks, and models by following the instructions outlined alongside the grey lines.

$\mathcal{C}_{Java}$, $\mathcal{C}_{Python}$, ... We define the natural language categories which contain all description/explanation/summarization of code etc.: $\mathcal{C}_{English}$, $\mathcal{C}_{Chinese}$, ... For simplicity, let's call them $\mathcal{C}_1$, $\mathcal{C}_2$, ...

In a category, there are infinitely many different objects. We name these objects $o_1$, $o_2$, ... Each object contains infinitely many programs, with *the same running outcome for each valid input*. We use this definition because we focus on the functional equivalence of code function. For example, one may have slightly different implementations of quick sorts, but they represent exactly the same function. [1]

Based on the definition of objects, we define the morphism between two objects, as "necessary edits to convert a function to another". In particular, we define the self-morphism for each object, as "edits that do not change the functionality of the program".

**Definition 2.2.** (Functor). A functor $F$ from a category $\mathcal{C}$ to a category $D$, written as $F : \mathcal{C} \rightarrow D$, maps an object $x \in \mathrm{Ob}(\mathcal{C})$ to $F(x) \in \mathrm{Ob}(D)$; as well as a morphism $f : x \rightarrow y$ in $\mathrm{Hom}(\mathcal{C})$ to $F(f) : F(x) \rightarrow F(y)$ in $D$, such that the following two properties hold:

- For every object $x$ in $\mathcal{C}$, $F(\mathrm{id}_x) = \mathrm{id}_{F(x)}$;

- For all morphisms $f : x \rightarrow y$ and $g : y \rightarrow z$, $F(g \circ f) = F(g) \circ F(f)$.

**Application 2.2.** (Functor of Mixture of Code and NL). We define the functor from $\mathcal{C}_1$ to $\mathcal{C}_2$, to be the transform from one language to another language, but with the same functionality. When it applies to two code categories, it usually represents code translation, and we define it as a "translation functor". When the functor is between a code category and a natural language category, it may have many possible meanings, for instance, we can define "generation functor" from NL to PL that generate code solution to a problem description, "explanation functor" from PL to NL that explains a piece of code, and "reproduction functor" from NL to PL that generates code snippets based on code descriptions.

## 2.2 Standardized Evaluation Platform

To show the practical use of categorical framework and support the community, we open-source our evaluation framwork[2]. Figure 3. shows the streamlined process of our standardized evaluation platform. Standardization within the framework is achieved through a focus on three critical aspects: data definition, task formulation, and APIs.

---

[1]We can also extend the definition of the object to be the set of code with not only the same outcome, but also with the same time or space complexity, but due to the difficulties of automatically evaluating the complexity, this can be extended for future work.

[2]The code can be found in https://github.com/scorpio-nova/CatCode.

**Data definition** The original datasets may consist of a mixture of code and natural languages, so a clear data definition is crucial. The code can be in different languages (Java, Python, JavaScript, etc.), and on different levels (single-line, function-level, file-level). Natural language can be in different relationships with the code (problem descriptions, code comments, demands, etc.). By establishing a standardized data definition, we ensure compatibility and facilitate the comparison and integration of different datasets and tasks. For implementation, we use a "data formatter" to rearrange code based on the definition of objects and categories. Additionally, we provide "predefined morphisms" and implement them based on `JavaTransformer`(Rabin et al., 2019) to automatically apply morphisms to code objects, which makes it easy for data augmentation.

**Task formulation** We use a categorical perspective to formulate diverse code-related tasks using the unified language of objects, morphisms, and functors. A good task formulation allows for a more generalized and flexible approach to defining more complex code problems, enabling the inclusion of a wide variety of code-related tasks and comparing their similarities and differences. Based on the task formulation, we carefully decide what parts of the code and NL should be the model input, ensuring a cleaner setting. Then we use "prompt selector" to select a suitable prompt for task description, and use "input constructor" to combine the data and prompt as task input.

**APIs** After the data is ready, we focus on the standardization of model APIs and test APIs. Standardized APIs promote transparency, fairness, and efficiency in the evaluation. By defining a clear and consistent set of APIs, we have integrated OpenAI models into the evaluation process and can test them using parallel requests. For postprocessing the model's output, we enable the extraction and filtering of plain text from the code-text mixture. For test APIs, we provide a "pairing test" API for evaluating the model's answer based on natural language, and an "execution test" API, which connects to `Mxeval`(Athiwaratkun et al., 2022) for compiling and running the tests for a given code.

## 3 Experiments

### 3.1 Use category framework for evaluation

In this section, we exhibit some experimental examples of how to use our platform. Meanwhile, we explore the following three types of tasks: discriminant task, generation task, and composite task, that correspond to those illustrated in Step 3 of Figure 1.

> **Discriminant task: Can the model exactly capture code meanings and identify similar/different code?**

(Related tasks: code detection, defect detection, code repair)

$\rightarrow$ Categorical perspective: Can the model identify the self-morphisms and other morphisms within the same code category?

> **Generation task: Can the model translate code between programming languages?**

(Related tasks: code translation)

$\rightarrow$ Categorical perspective: Can the model accurately perform code translation functor?

> **Composite task: Can the model reproduce the code after equivalent transformations?**

(Related tasks: code explanation, code summarization, code generation, code reproduction)

$\rightarrow$ Categorical perspective: Can the model preserve the functional equivalence after applying an explanation functor and then a reproduction functor?

We conduct three experiments from categorical aspects accordingly. For a common setting, we use three multilingual datasets: HumanEval-X(Zheng et al., 2023), MBXP(Athiwaratkun et al., 2022), and MathQA(Amini et al., 2019; Athiwaratkun et al., 2022) throughout the three experiments, and use five models: Text-Davinci-003, gpt3.5-turbo, Llama-2-7b-chat-

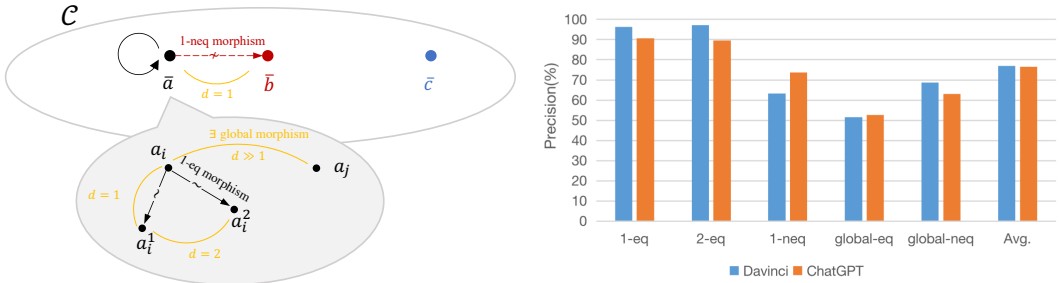

Figure 4: Morphism Identification Experiment. "1", "2" and "global" stands for the distance of the code. "Eq" and "neq" indicates whether the morphism is self-morphism. **(Left)** An illustration of morphisms and the definition of object distance. **(Right)** Comparison of Text-Davinci and ChatGPT for morphism identification.

hf, starchat-beta (Text-Davinci, ChatGPT, Llama2 and StarCoder for short accordingly) as common baseline models. For more detailed experimental settings and interesting results about specific model performance, please refer to Appendix A and B.

## 3.2 Experiment 1: Morphism Identification Within a Code Category

### 3.2.1 Categorical Perspective Settings

**Code Objects** We conduct the experiments within the PL category and define "the function with the same running outcome for every valid input" as the same object.

**Code Morphism** The general category perspectives only define morphisms between two code snippets, but morphisms do not have distance information. Considering fine-grained evaluation, we are curious about the model's ability to modify or debug the code, which corresponds to the local scale; and to write an equivalent new solution to a coding problem, which is of a global scale.

**Object Distance** To test the model on both local and global scales, we define the "distance" between two codes. As illustrated in Figure 4. In a PL Category $\mathcal{C}$, we use $\sim$ to express functional equivalence. $\bar{a} = \{x \mid x \in \mathcal{C}, x \sim a\}$ means the set of all code that has the same functionality as $a$. In this category, $\bar{a}$ and $\bar{c}$ are two different objects. Within $\bar{a}$, all code instances are functionally equivalent, e.g. $a_i$ and $a_j$. We can apply predefined morphisms to the objects. For example, we can apply 1-step equivalent morphisms(1-eq morphism) on $a_i$ to get new program instances still in $\bar{a}$, or apply 1-step nonequivalent morphism(1-neq morphism) to get new instance in another equivalent class $\bar{b}$. Distance $d$ represents the minimal number of morphisms needed to transform one program to another given a predefined morphism set. For example, Applying two different self-morphism on $a_i$ returns $a_i^1$ and $a_i^2$, then we need at least two steps to transform from $a_i^1$ to $a_i^2$. Those are local-scale equivalence. If it is hard to transform $a_i$ to $a_j$ within just a few steps(*e.g.*, 1 or 2 steps as illustrated here), we treat them as global-scale equivalence.

### 3.2.2 Implementation

**Dataset** For local morphism, we extract the Java functions within HumanEval-X, MBXP, and MathQA datasets. We perform the 9 local morphisms based on AST transformations.

1. `Variable Renaming (VR)`: rename a variable, with a new name randomly chosen from all variable names in the dataset, or use "$var_N$" for simplicity
2. `Boolean Exchange (BE)`: propagate the exchange of "true" and "false" throughout the method and preserve the same logic
3. `Loop Exchange (LE)`: exchange *for* loops and *while* loops
4. `Switch To If (SI)`: replace a switch statement with an equivalent if statement
5. `Unused Statement(US)`: insert an unused string declaration to a randomly selected basic block in a method

6. `Reorder Condition (RC)`: write the condition in reverse order (*e.g.*, change $i < n$ to $n > i$)
7. `Permute Statement(PS)`: swap two independent statements (i.e. without data or control dependence)
8. `Modify Condition(MC)`: change the binary operation in the condition (*e.g.*, change $i < n$ to $i \leq n$)
9. `Remove Else(RE)`: delete the else branch of the if-else statement

The first 7 morphisms are adopted from JavaTransform(Rabin et al., 2019) that conducts functionally equivalent transformations, which means they are self-morphisms. Other than self-morphisms, we add 2 morphisms: `ModifyCondition` and `RemoveElse`, that change the program's function.

For global morphisms, since the three datasets above do not contain multiple solutions to the same problem using the same PL, we complement with code from the test split of Code Contest(Li et al., 2022) dataset.

For dataset statistics and the detailed preprocess procedure, please refer to Table 2 in the Appendix.

**Models** We evaluate Text-Davinci and ChatGPT. We input (code, code) pairs from the datasets or generated by morphisms, and ask the model to answer whether they are functionally equivalent.

**Evaluation** We collect the model's yes/no responses and calculate the average precision for different morphisms.

### 3.2.3 Results

The results are demonstrated in Figure 4. Overall, Text-Davinci and ChatGPT make a tie. Locally, ChatGPT is better at identifying nonequivalent morphisms, and worse at equivalent ones. Globally, Text-Davinci is better at identifying global nonequivalent morphisms. Note that a random guess leads to a precision of 50%, so both models behave just slightly better than random guesses for identifying global equivalence.

## 3.3 Experiment 2: Translation Functor Between Different PL Categories

### 3.3.1 Categorical Perspective Settings

We investigate the model's ability to perform code translation by utilizing functors between two programming language categories. To isolate the influence of natural language, we exclude problem descriptions and code comments, forcing the models to focus solely on translating code from one programming language (PL) category to another. The input prompt may contain a request for translation using natural language (*e.g.*, "translate the below Java code to Python code").

### 3.3.2 Implementation

**Datasets** We conduct evaluations on the HumanEval-X, MathQA, and MBXP datasets. The input category is Java, while the output categories are JavaScript and Python.

**Models** We evaluate five models: Text-Davinci, ChatGPT, CodeGeeX, Llama2 and StarCoder. For CodeGeeX, code translation is supported inherently, so we simply input the Java code without additional text. For the other models, we provide prompts that request the model to do the translation.

**Evaluation** We extract the functions from the model's responses and assess their correctness using the Pass@1 rates of execution-based tests.

### 3.3.3 Results

The results are depicted in Figure 5. Among the three models, ChatGPT performs the best, exhibiting a slight advantage over Text-Davinci. With regards to the datasets, ChatGPT and

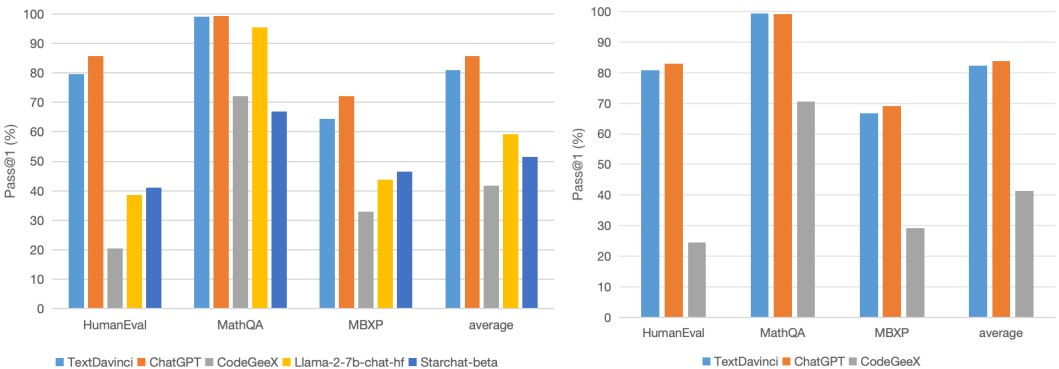

Figure 5: Comparison of performance for the model as translation functor between different PL categories. **(Left)** Translate from Java to JavaScript **(Right)** Translate from Java to Python.

Text-Davinci achieve nearly perfect translation accuracy on MathQA, whose data consists of functions with internal variable initialization, and without any input arguments, loops, or conditional statements. This indicates ChatGPT's and Text-Davinci's proficiency in reproducing exact numerical values and handling simple program structures.

### 3.4 Experiment 3: Explanation Functor and Reproduction Functor Between PL and NL Categories

#### 3.4.1 Categorical Perspective Settings

Code explanation and code generation tasks have been conducted separately in previous work. However, from a categorical perspective, we can make a combination. We define the explanation functor as the functor that maps an object from a PL category to an NL category, with a precise description of the code's function. We define the reproduction functor as the functor from an NL category to a PL category, that uses an explanation object in NL to reproduce the code of the same function in a PL category. If the reproduced code is functionally the same as the original code, we reckon the model preserves the functional equivalence after applying an explanation functor and then a generation functor.

#### 3.4.2 Implementation

**Datasets** We conduct the evalution on three datasets: HumanEval-X, MathQA, and MBXP. For consistency, we select Java as the programming language (PL) category for all evaluations.

**Models** We evaluate Text-Davinci, ChatGPT, Llama2 and StarCoder, prompting each model twice. In the first prompt, we ask the models to describe the code using natural language, including the precise function name, arguments, and return type, and provide sufficient information to reproduce the code. In the second prompt, we provide the model's explanation and ask it to translate the description back into code.

**Evaluation** To evaluate the correctness of the reproduced code, we extract the functions from the model's second responses and assess them using the Pass@1 rates of execution-based tests.

#### 3.4.3 Results

The results are depicted in Figure 6. On the same datasets and with the same two models, the average pass@1 rate is about 30% lower compared to that of the translation functor. This indicates the model has significant information loss in the explain-reproduce process.

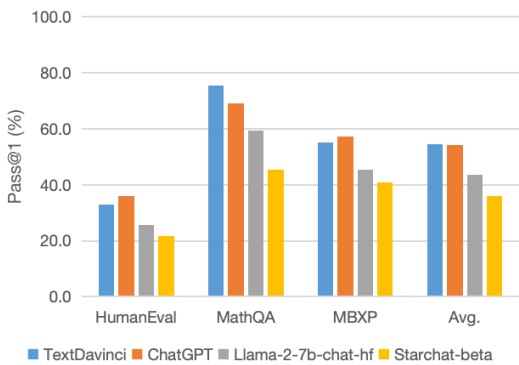

Figure 6: Comparison of model performance for the model as the combination of explanation functor and reproduction functor between PL and NL categories.

### 3.4.4 Additional Notes on the composite task

As models evolve, scores on high-quality datasets tend to saturate (HumanEval 90%). Should we abandon HumanEval, and call for massive efforts to construct new ones? We believe it's not the only way. With CatCode, we can easily define composite tasks, and make the evaluation effective again. The construction of composite tasks is mostly automatic, greatly reducing the labor for collecting new datasets.

Moreover, real-world applications often require model to handle those composite tasks. For example, developers may need to use the model to write new features, and synchronize code updates between Android and iOS versions. To this end, we propose and experiment on a new task "repair morphism + translation functor" using the CodeScope dataset. We use the code repair subset from the CodeScope dataset(Yan et al., 2023) and add a step for the model to translate from C++ to C after debugging C programs. Results on GPT-4 show that pass@1 drops from 26/60 in the original task to 18/60 in the composite task, which shows that the composite task is more difficult.

### 3.5 Summary

Based on our experiments, we have found that both Text-Davinci and ChatGPT models are capable of identifying the impact of one or two local morphisms on code function. However, identifying global morphisms proves to be challenging for these models. In terms of code translation, both ChatGPT and Text-Davinci models outperform other models. They demonstrate accurate reproduction of basic syntax, variable names, and numerical values. Nevertheless, when dealing with more intricate program structures and aligning data types across different programming language categories, these models encounter obstacles that hinder more precise translation.

It is important to note that maintaining functional equivalence between code and natural language presents a more formidable challenge compared to code translation.

## 4 Related Work

### 4.1 Coding Abilities of Large Language Models

Research on large language models trained on code has made significant strides. CodeBERT (Feng et al., 2020) and GraphCodeBERT (Guo et al., 2020) incorporated program Abstract Syntax Trees (AST) and data flow information to enhance code structure and context understanding. The Encoder-Decoder architecture in CodeT5 (Wang et al., 2021) and PLBART (Ahmad et al., 2021) supports multi-task learning, aiding code translation and generation. More recently, the emergence of Codex (Chen et al., 2021), ChatGPT, and StarCoder(Li et al., 2023) has brought GPT-based models into the mainstream for content generation tasks, including coding. These models have shown impressive performance across various domains,

but their proprietary nature, limited access to data, or lack of model checkpoints have made it challenging to thoroughly evaluate and understand their capabilities and limitations. Consequently, there is a need to develop a comprehensive framework to evaluate these black-box models.

## 4.2 Code Model Evaluation

Evaluation of code models is diverse, encompassing match-based, task-based, and execution-based frameworks. The match-based approach, exemplified by CodeBLEU(Ren et al., 2020), considers n-gram matches and AST similarity but may overlook functional equivalence in varying solutions. Task-based evaluations, like CodeXGLUE(Lu et al., 2021), offer comprehensive assessments across coding tasks but lack uniformity in data and task settings, limiting scalability. Execution-based evaluation, used in benchmarks like MBXP(Athiwaratkun et al., 2022) and MultiPL-E(Cassano et al., 2023), judges code correctness through test cases. Our work mainly focuses on providing a new framework for unifying the task-based evaluation. While our work also considers test cases from execution-based benchmarks, our focus extends beyond correctness evaluation to include code morphism, PL to NL functor aspects, and the composition of different tasks, for example the code reproduction task can be viewed as the combination of code explanation and generation tasks.

Moreover, code-related models are found to be sensitive to slight edits in the prompts, therefore to evaluate their robustness is crucial. Previous work applies semantics-preserving transformations on code docstrings and program, for better training and evaluation of models(Wang et al., 2022; Henkel et al., 2022). Apart from using transformations based on AST edits to ensure accuracy, we further investigate the models' robustness by providing a finer-grained characterization based on the edit distance of correct and incorrect edits.

## 5 Conclusion

Our contributions include introducing CatCode as a novel evaluation perspective based on category theory, which enables a comprehensive and mathematically abstract approach to evaluate LLMs that excel in understanding and generating a mixture of code and text. Based on categorical perspectives, we present a standardized automatic evaluation platform, which is adaptive to new datasets, tasks, and models. We evaluate competitive LLMs to provide valuable insights and find out current models' deficiency in recognizing functionally equivalent code and preserving information of code function between the code and its explanation. We open-source our platform, hoping to contribute to the comprehensive and standardized evaluation for LLMs by offering a categorical perspective to deal with the mixture of code and text. We hope that CatCode inspires further research and development in the field, leading to LLMs that can effectively assist developers in more complex coding tasks and contribute to advancements in artificial intelligence and software engineering as a whole.

## 6 Discussion

While we strive to apply categorical perspectives to offer a comprehensive and standardized way of evaluation, we find some limitations and room for future work. First, we use prompts to instruct the models to act as certain functors, but prompts that express similar meanings will lead to different results, we have not investigated which prompts are more suitable for the model or which models are more robust to different prompts.

In this paper, our experiments primarily focus on objects, morphisms, functors, and their combinations. For future work, more powerful concepts and properties in category theory can be used for more complex and composite coding tasks. For example, we can further investigate the setting of composing many morphisms and many functors, to further explore the commutative law and isomorphism properties of categories.

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

# Appendix

The appendix is divided into three main sections, each focusing on distinct aspects of the study. These sections are labeled as follows:

A. Implementation Details: This section covers the implementation process, including the GitHub link to our code, dataset details, data filtering, model specifications, and evaluation metrics, providing information on how to reproduce based on our standard evaluation pipeline.

B. Experimental Details: This part introduces the experimental settings and more experimental results in detail. We further discuss the model's performance on different morphism types in the identification experiment, and the common error types in the translation and reproduction experiments.

C. Case Study: This section offers some intuitive examples of the experiments. Through analysis of those examples, we can find some limitations of current models, prompts, and test scripts.

## A  Implementation Details

We have uploaded the code to GitHub: https://github.com/scorpio-nova/CatCode. A `README.md` file is provided, offering step-by-step instructions on how to use the evaluation platform and replicate the entire evaluation process in a standardized way.

In this section, we will provide additional details regarding the reproduction process, including the dataset split, data filtering, models used, and evaluation metrics.

### A.1  Dataset Details

The following datasets were used in our evaluation:

- HumanEval-X[3]: a human-crafted dataset with 164 problems, correct solutions in 5 different languages, and test cases. It is a benchmark for evaluating code generation and translation models. We use the Java split of HumanEval-X as input for applying morphism, code translation, and code explanation. Although `Mxeval` provide multilingual solutions, they are generated by models and do not make a distinction between correct and wrong solutions. In contrast, HumanEval-X provides correct solutions, making it a more suitable complement.

- MBXP: a benchmark for evaluating code generation models. It covers over 10 programming languages and is generated using a scalable conversion framework that transpiles prompts and test cases from the original Python datasets into the corresponding data in the target language. It's a dataset similar to HumanEval-X but does not ensure the correctness of the transpiled code.

- MathQA: a dataset for evaluating math problem solvers. All functions in this dataset have no input argument and return a number. The functions first initialize variables, then do some calculations, and do not use any loops or conditional statements. Compared to HumanEval-X and MBXP, the code in MathQA is simpler in structure and functionality.

- Code Contest: a large-scale competitive multi-language programming dataset used for training AlphaCode(Li et al., 2022). It consists of programming problems, test cases in the form of paired inputs and outputs, and both correct and incorrect solutions. Since the aforementioned datasets do not provide multiple solutions within the same programming language, Code Contest serves as a valuable complement by offering multiple solutions to the same problem, representing global equivalence and non-equivalence.

**Data filtering**

---

[3]https://huggingface.co/datasets/THUDM/humaneval-x

| Model | Morphism | Translation Functor | Explanation Functor | Reproduction Functor |
|-------|----------|---------------------|---------------------|----------------------|
| Davinci | ✓ | ✓ | ✓ | ✓ |
| ChatGPT | ✓ | ✓ | ✓ | ✓ |
| CodeGeeX | | ✓ | | |
| LLaMa2-7b | ✓ | ✓ | ✓ | ✓ |
| StarCoder | ✓ | ✓ | ✓ | ✓ |

Table 1: Baseline Models and their supported evaluation tasks.

- Only the Java split of HumanEval-X, MBXP, and MathQA datasets are used. The local morphism is applied by `JavaTransformer` by AST edits at the function level. To use JavaTransformer, code must compile successfully and contain a single function. We filter out code in the `Mxeval` datasets that are generated by the model and cannot compile or contain multiple functions.

- Due to the large size of the Code Contest dataset, only the test set is used. Considering that the maximum token length of Text-Davinci is 2048, we filter out solutions with a length greater than 500. This ensures that the code pair length remains $\leq 1000$, allowing for additional tokens for prompt content and response. After filtering, 97 problems remain, and a problem may have multiple correct and incorrect solutions in different languages. We utilize the Java and Python splits of this dataset.

### A.2 Model details

We utilize the following baseline models by making API calls to their official endpoints: Text-Davinci-003 (referred to as Davinci), ChatGPT, CodeGeeX. The default hyperparameters of these models are used, with the exception of setting `max_token` to 500 when an early stop of the answer is observed.

Table 1 provides an overview of the tasks supported by each model. Since Text-Davinci-003 and ChatGPT support various types of text inputs and outputs, they are capable of performing all tasks. On the other hand, the CodeGeeX API does not support natural language output, so we only employ it in translation functor experiments.

### A.3 Evaluation Metrics

To assess the performance of the model, we conduct two types of tests: pairing test and execution-based test.

The pairing test is utilized for morphism identification. The task involves comparing two code snippets and requesting the model to explicitly answer "True" or "False", along with providing a comparison. For automatic evaluation, we extract the True" or "False" answer. The precision score is used to evaluate the model's ability to identify different equivalence classes. For equivalent objects, the precision is calculated as $Precision(eq) = \frac{TP}{TP+FN}$, while for nonequivalent objects, the precision is calculated as $Precision(neq) = \frac{TN}{TN+FP}$.

The execution-based test is employed for both translation and reproduction experiments. We extract the functions from the model's responses and evaluate their correctness using the Pass@1 rates of execution-based tests. This test ensures that the translated/reproduced code produces the same expected results as the original code.

## B  Experimental Details

### B.1  Morphism Identification

**Data statistics**. Table 2 provides the statistics of the raw data, filtered data, and constructed pairs for the morphism identification experiment. The filtering strategy, described in Section A.1, involves removing comments and docstrings, resulting in filtered data consisting

| Dataset | #after filter / #raw data | #equivalent pair | #nonequivalent pair |
|---|---|---|---|
| Humaneval-X | 159 / 164 | 477 | 173 |
| MBXP | 953 / 974 | 2849 | 1010 |
| MathQA | 1734 / 1881 | 4852 | 1418 |
| Code Contest | 97 / 164 | 97 | 366 |

Table 2: Dataset statistics for morphism identification experiment.

| HumanEval | MathQA | MBXP |
|---|---|---|
| BE-VR (66.67) | MC (45.66) | BE (33.33) |
| US (74.74) | RC-US (85.71) | PS-US (69.57) |
| LE-RC (78.57) | PS-US (87.84) | RC-US (70.39) |

Table 3: Morphism types with the three worst average precision scores of each dataset. As an example for notations, BE-VR (66.67) stands for a 2-eq morphism of "Boolean Exchange" and "Variable Renaming" with a precision score of 66.67%.

solely of Java code. The constructed pairs are generated through sampling. It is worth noting that certain morphisms can have multiple outputs for a given input. For instance, the `Variable Renaming` morphism can be applied to all variables in a function, resulting in multiple outputs equal to the number of variables. To balance the number of outputs across different morphism types, random sampling is employed when there are more than two applicable morphism types. For each code sample, two types of morphisms are first sampled, followed by sampling one output for each morphism type.

**Discussion of different morphism types.**

Which morphism types are harder to be identified, and are they hard across datasets? We calculate the average precision scores for 1-eq, 2-eq and 1-neq local morphisms for ChatGPT answers, as shown in Table 3.

The following observations can be made:

- For one morphism: "Unused Statements", "Modify Condition" and "Boolean Exchange" are particularly difficult to identify. One possible reason for this difficulty is that these morphisms involve subtle changes or transformations in the code that may not be easily recognizable based on the surrounding context alone. It requires a more nuanced understanding of the code logic and structure to detect these morphisms

- For 2-eq morphisms: The observation that "Unused Statements" remains challenging when combined with other morphisms suggests that the presence of multiple morphisms in the code can further complicate the identification task.

- Dataset-specific difficulties: The results indicate that the difficulty of identifying "Unused Statements" persists across datasets, suggesting that this morphism type poses inherent challenges in code comprehension. On the other hand, the specific difficulty in identifying "Modify Condition" in the MathQA dataset may be attributed to the nature of the dataset itself, which focuses on math problem solvers.

## B.2 Translation Functor

**Data statistics**. We use the filtered data in Table 2 as input, i.e. 159, 953, and 97 Java Snippets for HumanEval-X, MBXP, and MathQA, respectively. Each model generates a Python object and a JavaScript object for each Java object. During test phase, since the `Mexeval` dataset is missing 3 test cases in the Java split of HumanEval (TaskID: 32, 38, 50), we use 156, 953, and 97 Java functions to calculate the pass@1 rates for HumanEval-X, MBXP, and MathQA respectively.

**Results statistics**. The pass@1 score of different models on different datasets is shown in Table 4. Overall, ChatGPT has a relatively higher translation ability.

| Model | HumanEval | | MathQA | | MBXP | | Avg. |
|---|---|---|---|---|---|---|---|
| | Java->Py | Java->JS | Java->Py | Java->JS | Java->Py | Java->JS | |
| Text-Davinci | 80.8 | 79.5 | **99.4** | 99.1 | 66.6 | 64.4 | 81.6 |
| ChatGPT | **83.0** | **85.8** | 99.2 | **99.4** | **69.0** | **72.1** | **84.8** |
| CodeGeeX | 24.5 | 20.5 | 70.5 | 72.2 | 29.2 | 32.8 | 41.6 |

Table 4: Translation pass@1 score (%) statistics, corresponding to the Figure 5 **(Left)**. The models translate Java code to two languages Python (Py) and JavaScript (JS), on three datasets. The average (Avg.) is the average score over the six scores.

| Model | HumanEval | | MathQA | | MBXP | |
|---|---|---|---|---|---|---|
| | Java->Py | Java->JS | Java->Py | Java->JS | Java->Py | Java->JS |
| Davinci | Unknown (10) | ReferenceErr. (24) | Unknown (9) | **Test Failure (10)** | Unknown(175) | ReferenceErr. (285) |
| | NameErr. (7) | **Test Failure (18)** | IndentationErr. (1) | ReferenceErr. (2) | NameErr. (47) | **Test Failure (174)** |
| | **Test Failure (2)** | TypeErr. (5) | NameErr. (1) | SyntaxErr. (2) | ReferenceErr. (26) | TypeErr. (41) |
| ChatGPT | Unknown (14) | ReferenceErr. (21) | Unknown (8) | **Test Failure (9)** | Unknown (145) | ReferenceErr. (247) |
| | NameErr. (6) | **Test Failure (17)** | SyntaxErr. (4) | SyntaxErr. (1) | NameErr. (81) | **Test Failure (101)** |
| | SyntaxErr. (2) | SyntaxErr. (1) | NameErr. (1) | \ (0) | SyntaxErr. (15) | SyntaxErr. (8) |
| CodeGeeX | Unknown (50) | **Test Failure (82)** | ReferenceErr. (330) | **Test Failure (394)** | Unknown (175) | ReferenceErr. (559) |
| | NameErr. (31) | ReferenceErr. (26) | NameErr. (107) | ReferenceErr. (75) | NameErr. (47) | **Test Failure (289)** |
| | **Test Failure (10)** | TypeErr. (6) | Unknown (38) | SyntaxErr. (11) | ReferenceErr. (26) | TypeErr. (55) |

Table 5: Most frequent failure types of translation functor. The data in the table cell is written as ErrorType (frequency), and Err. is short for Error. Specifically, the "Unknown" type only exists in Python and "Test Failure" represents the code that complies successfully but fails to pass some of the test cases. Other types of errors in this table are primarily caused by compilation errors.

**Discussion of failure types.** Table 5 represents the most frequent failure types of translation functor on three datasets. We observe that most errors are compilation errors rather than failures at corner test cases.

These errors are often caused by type mismatches between different programming languages. For instance, when translating a list in Java, there are multiple potential types in Python, including list, tuple, and List from the typing module.

The occurrence of "NameError" and "ReferenceError" indicates an unfaithful translation of variable names by the model or mismatched function input arguments across different languages for the same problem. This is a limitation inherent in the current evaluation setup.

Overall, the models have demonstrated the ability to perform correct translations in most cases between different programming languages. The errors mainly arise from type alignment issues between programming languages and local variable definitions. These errors are typically easy to debug for programmers if provided with relevant information. Future work could involve passing error messages to the model and allowing it to debug itself.

## B.3 Explanation and Reproduction Functor

**Data statistics**. Same as Appendix B.2, we use 156, 953, and 97 Java functions to calculate the pass@1 rates for HumanEval-X, MBXP, and MathQA separately.

**Results statistics**. Table 6 shows the pass@1 rates of different models. TextDavinci is better at MathQA, while ChatGPT is better at HumanEval and MBXP datasets. The pass@1 score is significantly lower compared to translation functor experiments, indicating that it's a more difficult task.

**Discussion of failure types.** Table 7 shows the most frequent failure types of translation functor on the three datasets. Based on the results, it appears that both models encounter similar types of failures across the different evaluation tasks. The "Test Failure" type is particularly prominent, indicating that the reproduced code, although compiling successfully,

| Model | HumanEval | MathQA | MBXP | Avg. |
|---|---|---|---|---|
| TextDavinci | 33.0 | **75.5** | 55.2 | **54.6** |
| ChatGPT | **35.9** | 69.1 | **57.1** | 54.0 |
| Llama-2-7b-chat-hf | 25.9 | 59.4 | 45.4 | 43.5 |
| Starchat-beta | 21.8 | 45.4 | 40.7 | 36.0 |

Table 6: Pass@1 score (%) statistics of the reproduced code, corresponding to the Figure 6 **(Right)**.

| Model | HumanEval | MathQA | MBXP |
|---|---|---|---|
| | CannotFindSymbol (47) | **Test Failure (350)** | **Test Failure (181)** |
| TextDavinci | TypeError (38) | UninitializedVariable (24) | CannotFindSymbol (164) |
| | **Test Failure (22)** | CannotFindSymbol (18) | TypeError (27) |
| | CannotFindSymbol (46) | **Test Failure (401)** | CannotFindSymbol (137) |
| ChatGPT | TypeError (39) | DuplicateDeclaration (60) | **Test Failure (126)** |
| | **Test Failure (17)** | SyntaxError (44) | UnexpectedEOF (70) |

Table 7: Most frequent failure types of the reproduced code. The data in the table cell is written as ErrorType(frequency). Specifically, the "Test Failure" type represents the code that complies successfully but fails to pass some of the test cases. Other types of errors in this table are primarily caused by compilation errors.

fails to pass some of the test cases. Additionally, both models also encounter errors related to symbol identification, such as "CannotFindSymbol" and "TypeError". These errors are primarily compilation errors.

It's worth further investigating the causes behind these failure types and exploring strategies to improve the models' performance, especially in handling test cases and resolving symbol-related errors.

## C   Case Study

In this Section, we analyze some classical cases from our three experiments.

### C.1   Morphism Identification

Refer to Figure 7 and Figure 8. Those cases show the model can identify the literate differences between the code pairs but fails to clearly interpret their functional difference.

### C.2   Translation Functor

According to the previous statistics, most translation errors happen in the compilation stage. We are also concerned about if there are no compile errors, when will the model make mistakes. Figure 9 shows such a case where some information about the numerical data type is missing when translation.

### C.3   Explanation Functor and Reproduction Functor

Figure 10 shows a case where information loss and hallucination happen during explanation and reproduction. This is often the case when the model deals with the MathQA dataset. To improve the performance of the model, using few-shot prompts to illustrate how to explain the code may be useful.

**Prompt Text:**

Do the two codes below have the same output for each input? In your first sentence, answer "True" or "False". Then points out all the differences of the two codes.

| HumanEval/154 | **LoopExchange**-ReorderCondition |
|---|---|

```
public boolean cycpatternCheck(String a, String b) {
    int l = b.length();
    String pat = b + b;
    for (int i = 0; i <= a.length() - l; i++) {
        for (int j = 0; j <= l; j++) {
            if (a.substring(i, l + i).equals(pat.substring(j, j
+ l))) {
                return true;
            }
        }
    }
    return false;
}
```

```
public boolean cycpatternCheck(String a, String b) {
    int l = b.length();
    String pat = b + b;
    for (int i = 0; i <= a.length() - l; i++) {
        {
            int j = 0;
            while (j <= l) {
                if (a.substring(i, i + l).equals(pat.
substring(j, j + l))) {
                    return true;
                }
                j++;
            }
        }
    }
    return false;
}
```

**ChatGPT Response**

False. Differences:

- In code A, the while loop is used to iterate through the `l+1` possible subpatterns of `a` that have length `l`. In code B, the for loop is used instead.

- In code B, the condition for the for loop is `j <= l`, which means it checks if `j` is less than or equal to `l`. This will result in the last possible subpattern being ignored because `j` will stop at `l` and not check anything after it. Therefore, the correct condition should be `j < l`.

- In code B, the condition for comparing the substrings is `if (a.substring(i, l + i).equals(pat.substring(j, j + l)))`. This checks if the substring of `a` starting from index `i` with a length of `l` equals the corresponding substring of `pat`. In code A, the condition is similar, except it uses `a.substring(i, i + l)` instead. Both conditions are correct, but they differ slightly in their expression.

| HumanEval/129 | LoopExchange-**ReorderCondition** |
|---|---|

**ChatGPT Response**

False. The codes have differences in their inner loops and conditions inside the loops.

Code A uses a while loop to iterate through the columns of each row, whereas Code B uses two for loops to iterate through both rows and columns of the grid.

Code A also checks whether a cell contains a 1 before checking its neighbors, while Code B simply iterates through every cell regardless of its value.

Finally, there is an error in Code B where it checks the neighbor to the bottom incorrectly: it should use `i + 1` instead of `1 + i`.

Figure 7: Two failed cases of ChatGPT in the morphism identification experiment. For "HumanEval/154", ChatGPT fails to interpret the loop execution condition correctly. For "HumanEval/129" ChatGPT reckon equivalent math expression as different. To improve the model's reasoning ability, a possible way is to utilize the method of Chain of Thoughts(CoT) to construct better prompts and give the model some time to think before making a decision.

---

**Prompt Text:**

Do the two codes below have the same output for each input? In your first sentence, answer "True" or "False". Then points out all the differences of the two codes.

| HumanEval/94 | BooleanExchange-VariableRenaming |

```
public int skjkasdkd(List<Integer> lst) {

    int maxx = 0;

    for (int i : lst) {

        if (i > maxx) {

            boolean isPrime = !(i != 1);

            for (int j = 2; j < Math.sqrt(i) + 1; j++) {

                if (i % j == 0) {

                    isPrime = true;

                    break;

                }

            }

            if (!(isPrime)) {

                maxx = i;

            }

        }

    }

    ...

    return sum;

}
```

```
public int skjkasdkd(List<Integer> var) {

    int maxx = 0;

    for (int i : var) {

        if (i > maxx) {

            boolean isPrime = i != 1;

            for (int j = 2; j < Math.sqrt(i) + 1; j++) {

                if (i % j == 0) {

                    isPrime = false;

                    break;

                }

            }

            if (isPrime) {

                maxx = i;

            }

        }

    }

    ...

    return sum;

}
```

**ChatGPT Response**

False.

Differences:

- In code A, the `isPrime` boolean variable is assigned to `!(i != 1)` which is equivalent to `i == 1`. Whereas in code B, it is assigned to `i != 1`.

- In code A, the condition for updating the `maxx` variable is `if (!(isPrime))` which means if the number is not prime. Whereas in code B, it is `if (isPrime)` which means if the number is prime.

- ...

- Inside the inner loop of the `isPrime` check, code A sets `isPrime` to `true` if a factor of `i` is found and breaks out of the loop. This is incorrect since it sets `isPrime` to true even when a factor is found, whereas it should set it to false because `i` has factors other than 1 and itself. Code B correctly sets `isPrime` to `false` in this case.

- ...

Figure 8: A failed case of ChatGPT in the morphism identification experiment. ChatGPT understands the meaning of the word "isPrime" and points out the logical error. Although logical error exists in natural language, the code function is still the same. ChatGPT fails to find that code A is still equivalent to B.

**Prompt Text:**

Translate the below *Java* code to *JavaScript* code. The function header is *function isPerfectSquare(n) { {*. You should only answer the *JavaScript* function body without further comments and explanations.

**MBXP/803**

| Java Code (from the dataset) | JavaScript Code (by TextDavinci) |
|---|---|

```
public static Boolean isPerfectSquare(int n) {
    if (n < 0) {
        return false;
    }
    int sqrt = (int) Math.sqrt(n);
    return sqrt * sqrt == n;
}
```

```
    if (n < 0) {
        return false;
    }
    let sqrt = Math.sqrt(n);
    return sqrt * sqrt == n;
}
```

**Test Cases**

```
var arg00 = 10;
var x0 = isPerfectSquare(arg00);
var v0 = false;
if(v0)){
    throw 'Error at 1th assert statement. Value = ' + JSON.stringify(x)
}
var arg10 = 36;
var x1 = isPerfectSquare(arg10);
var v1 = true;
if(!compare(x1, v1)){
    throw 'Error at 2th assert statement. Value = ' + JSON.stringify(x)
}
var arg20 = 14;
var x2 = isPerfectSquare(arg20);
var v2 = false;
if(!compare(x2, v2)){
    throw 'Error at 3th assert statement. Value = ' + JSON.stringify(x)
}
```

**Execution Results**

```
...
throw 'Error at 3th assert statement.
```

Figure 9: A notable failure case occurred during the translation functor experiment with Text-Davinci. Text-Davinci fails to consider data type conversions during translation. This issue becomes evident when examining the three test cases provided in Mxeval, as the execution encounters an error in the third case. However, if it were not for the incorrect test scripts generated by the model in Mxeval, the translated code should have been identified as incorrect in the first case. This particular case highlights the importance of enhancing the correctness of model-generated execution test scripts for a more accurate test of code.

**Prompt Text:**

(Explanation task) Please describe the following code written in ``` using natural language. Your description should include the exact function name, its arguments, and the return type. Additionally, provide enough details such as variable initialization to allow someone to reproduce the code based on your explanation.

(Reproduction task) Translate the below code description to Java code: <Explanation>

**MathQA/1014**

| Java Code (from the dataset) | Reproduced Code (by Text-Davinci) |
|---|---|
| ```
public static double problem() {
    double n0 = 2.0;
    double n1 = 7.0;
    double n2 = 57.0;
    double n3 = 3.0;
    double n4 = 6.0;
    double n5 = 63.0;
    double n6 = 5.0;
    double n7 = 9.0;
    double n8 = 206.0;
    double n9 = 5.0;
    double n10 = 8.0;
    double t0 = Math.pow(n9, Math.min(n3, 5));
    double t1 = Math.pow(n10, Math.min(n0, 5));
    double answer = t0 + t1;
    return answer;
    }
``` | double n0 = 2;
double n1 = 4;
double n2 = 1;
double n3 = 3;
double n4 = 2;
double n5 = 8;
double n6 = 10;
double n7 = 3;
double n8 = 14;
double n9 = 2;
double n10 = 3;
double t0 = Math.pow(n9, n10);
 double t1 = Math.min(n3, n0);
 return t0 + t1; |

**Explanation (by Text-Davinci)**

This is a public static method called 'problem()' that does not take any arguments and returns a double value. The method initializes the variables n0 to n10 with double values and assigns them to the specified numbers. It then uses the Math.pow() method to use the variables n9 and n10 as the base for the power and the Math.min() method to use the variables n3 or n0 as the second parameter. It then assigns these two values, t0 and t1, as addends of a sum and returns the answer, a double value.

**Execution Results**

Exception -- test case 0 did not pass.

Figure 10: A failure happen in the explanation and reproduction functor experiment. Text-Davinci doesn't explain the code with any numerical details, and it makes up some numbers during reproduction.

