# OpenReview forum: "CatCode: A Comprehensive Evaluation Framework for LLMs On the Mixture of Code and Text"
_colmweb.org/COLM/2024/Conference — COLM_

### Official Review · Reviewer_VBwZ · 2024-05-11

**Rating:** 7
**Confidence:** 3
**Ethics Flag:** 1

**Summary:**

This paper proposes an automated evaluation framework called CatCode, designed to assess the capabilities of various large-scale language models, such as ChatGPT, in solving programming problems. The framework leverages category theory as a mathematical abstraction to evaluate code-related tasks and introduces concepts such as code debugging and transformation, code translation, code generation, and interpretation. This approach is comprehensive and standardized, allowing for the evaluation of mixed-code tasks of different types.

In terms of quality, the paper provides clear definitions and detailed explanations, enabling readers to comprehend and apply the CatCode framework.

In terms of originality, the paper introduces a novel evaluation framework that addresses limitations within existing evaluation methods by utilizing the principles of category theory.

**Reasons To Accept:**

Provides a comprehensive and standardized evaluation framework, facilitating a better understanding of the capabilities of large-scale language models in solving programming problems.

Utilizes the principles of category theory to address limitations in existing evaluation methods, showcasing innovation.

**Reasons To Reject:**

Relies only on existing small datasets for validation, requiring further experimental validation and refinement of the framework, such as in the case of Codescope.

The framework introduces a difference in the Morphism Identification task compared to previous mainstream code evaluation tasks, while the other two tasks remain consistent or a combination of previous tasks.

---

> ### Author Rebuttal · Authors · 2024-05-31
>
> Thank you for acknowledging the quality, originality, comprehensiveness, and standardness of our work.
> ### Q1: Relies on small datasets ... Further experimental validation ... Such as in the case of CodeScope
> In response to the reviewer's concerns, we conduct a "code-repair morphism+translation functor" task on the CodeScope dataset.
> - We used the `code_repair` subset and added a step for the model to translate C++ ->C after debugging. Results on `GPT-4` show that **pass@1 drops** from 26/60 in C++ to 18/60 in C. The task is more complex and has real-world applications, such as synchronizing code updates between Android and iOS versions.
>
> **Dataset Justification:**
> -  Dataset Size: Our Morphism Identification task uses ~10k test samples, and other tasks use ~2k samples each, which are substantial for code evaluation, comparable to CodeScope's average of ~1.7k samples/task.
> -  Framework Advantage: Our framework enables the augmentation of datasets by morphisms and the composition of functors. This can make datasets larger, apply to new tasks (Sec3.3.2), and new composite tasks (Sec3.3.4).
> ### Q2: Integration and Innovation in Task Design
> We appreciate the positive feedback on our new morphism identification, and provide further explanation for the other two:
> - **Translation functor task**: It shows how traditional tasks fit within this framework. The advantage of our framework is that it **embraces all mainstream coding tasks** :) instead of excluding or replacing them. This supports further composition with other morphisms and functors (e.g. The CodeScope task above).
> - **Composite functor task**: Combining various coding tasks is **Non-trivial**, because previous evaluation benchmarks use different datasets, different data fields, and independent evaluation pipelines for each task (e.g.CodeXGLUE). Our framework has the advantage of **systematically and automatically enabling task composition** (Please refer to Reviewer#kBTR_Q1 for details).
>
> Category Theory Framework Advantages in task design:
> - **Math abstraction**: Offers unified data definitions for various coding tasks, providing a conceptual basis for evaluating code performance.
> - **Comprehensive perspective**: Many studies have found that models pre-trained on code (e.g.DeepSeek-Coder, CodeLlama) perform better on mathematics. Coding tasks also have inter-relations. The category theory framework can serve as a good **starting point for understanding the correlations among various coding tasks**.

---

> > ### Comment · Reviewer_VBwZ · 2024-06-03
> >
> > The author addressed some of my concerns, and I am willing to increase the score from 6 to 7.

---

> > > ### Author Response · Authors · 2024-06-04
> > >
> > > Dear Reviewer,
> > >
> > > Thank you for acknowledging the response and for considering an increase in your evaluation score! We deeply appreciate your valuable feedback and the opportunity to refine our work further.
> > >
> > > We are committed to ensuring the highest quality of our research and welcome any additional suggestions you might have that could further improve our paper.
> > >
> > > Thank you once again for your constructive critique and your acknowledgment of our work :)

---

### Official Review · Reviewer_kBTR · 2024-05-12

**Rating:** 4
**Confidence:** 3
**Ethics Flag:** 1

**Summary:**

This paper discusses methods to evaluate the abilities of LLMs in solving coding problems that involve a combination of code and text. The authors introduce a novel evaluation perspective for code-related tasks. The key idea is to incorporate category theory, which aims to provide a comprehensive and mathematically abstract framework to evaluate LLMs, encompassing a wide range of code-related tasks. An advantage of the proposed framework is its adaptability to new datasets, tasks, and models, making it a potential standardized automatic evaluation platform. This paper evaluates competitive LLMs to identify current models' deficiencies in recognizing functionally equivalent code and preserving information about code function between the code and its explanation.

**Questions To Authors:**

* In Section 3.2.2, __"For local morphism, we extract the Java functions within HumanEval-X, MBXP, and MathQA datasets. We perform the 9 local morphisms based on AST transformations."__ and __"For global morphisms, since the three datasets above do not contain multiple solutions to the same problem using the same PL, we complement with code from the test split of Code Contest(Li et al., 2022) dataset."__
If I am not mistaken, it does not seem to provide clear numbers of questions for the local morphism task in the three datasets. This is also the case for the global morphism task conducted in this paper. Please provide more details on the evaluation settings for Section 3.2 so that readers can reproduce the experiments.


* Three different experimental results, namely Morphism Identification, Translation Functor, and Explanation Functor, conducted in this paper were presented in Sections 3.2.3, 3.3.3, and 3.4.3. However, the benefit of using the proposed method, specifically incorporating category theory, is not clearly described, particularly for Sections 3.3.3 and 3.4.3. Could you explain how readers can interpret the benefit of the proposed method from these results?

* In conclusion, the authors mention, __"Based on categorical perspectives, we present a standardized automatic evaluation platform, which is adaptive to new datasets, tasks, and models."__  However, I am not sure if this paper clearly demonstrates the ability to adapt to new tasks, as it mainly shows the adaptation ability for datasets and models.

**Reasons To Accept:**

* This paper tackles an important evaluation aspect for code-related tasks.
* This paper introduces a novel evaluation method for code-related tasks based on the category theory.

**Reasons To Reject:**

* The idea of incorporating category theory is interesting and unique. However, the merit of incorporating category theory for the evaluation was not strongly supported by the experiments (if my understanding is correct. See the questions to authors section. I am willing to increase my score if I have misunderstood and the authors convince me in the rebuttal.)

* The authors may need to show results of simple baseline methods without applying the proposed method and compare the results under fair conditions. Otherwise, it is hard to understand the actual benefit of using the proposed method.

---

> ### Author Rebuttal · Authors · 2024-05-31
>
> Thank you for recognizing our work's importance and novelty. Your queries help refine our approach.
> ### Q1: The benefit of the method & ability to adapt to new tasks
> The key benefit of our method is its capacity to define novel, challenging tasks, and adapt existing datasets to the tasks without human labor.
> ### From the experiments:
> 1. Sec3.3.2: **Challenging New Task**
>     - Inspiration: A fundamental question of category theory is object relationships, leading to natural thinking of the task.
>     - Key results: Experiment 1 **reveals a significant gap in models' lexical versus functional understanding**, (even advanced models like ChatGPT show ~50% accuracy at global morphism, just random guessing).
> 2. Sec3.3.4: **Composite Task**
>     - Relation to Category Theory: The unified math abstraction, and the associativity property of Morphisms and functors help us define composite tasks.
>     - Key insights for future work: As models evolve, scores on high-quality datasets tend to saturate (HumanEval~90%). Should we abandon HumanEval, and call for massive efforts to construct new ones? We believe it's not the only way! **With CatCode, we can easily define composite tasks, and make the evaluation effective again. The construction of composite tasks is mostly automatic, greatly reducing the labor for collecting new datasets** (and may prevent test set contamination).
> ### Potentials for Automatically Adapting New Tasks
> 1. **New composite task**: We propose and experiment on a new task "repair morphism + translation functor" using the CodeScope dataset. (Ref:Reviewer#VBwZ-Q1)
>     - Practical Application: This task can automate cross-platform consistency, such as synchronizing bug fixes or features between Android and iOS versions of an app.
> 2. **Future potential tasks**
>     - Other morphism+translation functor (harder than pure morphism/translation, run execution to eval)
>     - Self-morphism+summarization functor (produce many summarizations with the same meaning, can be judged by GPT without human-written ground truth), etc. We'd open-source the tool for transforming single tasks into composite tasks.
> ### Q2: No clear numbers of questions for the local morphism task in the three datasets... details to reproduce the experiments
> Please refer to Table 2 in Appendix A for the number of questions. In the appendix, we provide (1) link to anonymized code (2) 5 pages in Appendices A&B detailing the reproduction steps. We'll add more details in our revised version.

---

> > ### Author Response · Authors · 2024-06-07
> > **More details on the questions concerned**
> >
> > Dear reviewer,
> >
> > We would like to first thank you again for your constructive comments and valuable suggestions! Since we are nearly at the end of the discussion phase, we wish to clarify and expand on a few points based on your queries.
> >
> > 1. Regarding question 1 on the experimental details for the morphism identification task:
> >
> >    - Numbers of questions for the morphism identification task
> >
> >      (1) Local morphism task of the three datasets:
> >
> >      | question type  | Humaneval | MBXP | MathQA |
> >      | ----- | --------- | ---- | ------ |
> >      | 1-eq  | 318       | 1896 | 3237   |
> >      | 2-eq  | 159       | 953  | 1615   |
> >      | 1-neq | 173       | 1010 | 1418   |
> >      | total | 650       | 3859 | 6270   |
> >
> >      (2) For the global morphism task in the `CodeContest` dataset, there are 97 #global-eq and 366 #global-neq questions.
> >
> >      (3) In Appendix A, we provide more details about processing the question for fair evaluation(e.g.removing code comments that may leak some information for identification), and the filtering policy to make the number of different morphism types more balanced.
> >
> >    - More reproduction details: In the appendix, we provide (1) a link to anonymized code and (2) 5 pages in Appendices A&B detailing the reproduction steps and experimental statistics. We acknowledge that the main text was less detailed which may let you confused. We are also willing to update the main text with those important data statistics and reproducibility-related information as you suggested :)
> >
> > 2. For question 2: We want to point out that CatCode is the first paper to highlight the "task unification" and "task composition" in the field of code-related LLM evaluation. The benefits of the CatCode are that
> >
> >    - **For LLMs**: The paper underscores the importance of recognizing functional equivalence under varying lexical similarities—a critical challenge in coding. It also illustrates how LLMs might struggle with composite tasks despite performing well on singular tasks.
> >    - **For new tasks**: As elaborated in the Rebuttal, category theory provides the **capacity to define new challenging tasks** E.g. morphism-identification, composite code reproduction task in the paper; debug+translate task for CodeScope; and can support more new tasks in the future (Please refer to rebuttal for Q1 for detailed elaboration and more examples).
> >    - **For existing datasets**: The unified math abstraction and standard pipeline can **adapt existing datasets to those new tasks automatically and systematically**. This can alleviate high-quality datasets from saturation, and reduce human labor to create new datasets for new tasks.
> >    - The above benefits are unique contributions to supplement previous evaluations, specifically there are the case for two classical previous work
> >
> >        1. CodeXGLUE (Shuai Lu *et al.*, 2021) offers a comprehensive benchmark. From its leaderboard (https://microsoft.github.io/CodeXGLUE/), we can see researchers following on single tasks, but there's currently no following work that conducts all the tasks to obtain the overall performance. It's partly due to the complexity brought by the lack of standardness: they use almost completely different datasets, data formats, and pipelines for different tasks (ref: the first table in https://github.com/microsoft/CodeXGLUE). **Our work addresses this by introducing unified task definitions and a standard pipeline, making it easier for researchers to follow.**
> >
> >        2. CodeScope(W Yan *et al.*, 2024) is more standard in the model API and data format, but they don't further build the connection among coding tasks or include any composite tasks (https://github.com/WeixiangYAN/CodeScope). With people increasingly relying on LLMs to conduct composite coding tasks (e.g. We may ask LLMs to generate the code, debug the code, and write docstrings in the same dialog), and we've found by experiments that the composite tasks are hard. Therefore, **the category framework fills this gap, facilitating composite task definition and adapting datasets to them in an automatic way.**
> >
> >       In summary, our contributions are **both theoretical**—providing a unified mathematical abstraction for understanding the relationships among coding tasks—**and practical**, by standardizing the evaluation pipeline to enable automatic and systematic composite task assessments.
> >
> > Again, we sincerely appreciate your insightful questions and feedback, which certainly help refine our work on quality and clarity. We hope our responses adequately address your queries, and are eager to address any further questions or concerns you may have :)

---

### Official Review · Reviewer_hN4m · 2024-05-12

**Rating:** 7
**Confidence:** 4
**Ethics Flag:** 1

**Summary:**

In this paper, the authors introduce a novel evaluation framework called CatCode, which is based on category theory principles. CatCode offers a comprehensive and mathematically abstract approach to evaluating the performance of large language models (LLMs) in tasks involving the understanding and generation of a mixture of code and natural language text.

One of the key contributions of this work is the development of a standardized automatic evaluation platform that is adaptive to new datasets, tasks, and models. The authors evaluate ChatGPT, Text-Davinci and CodeGeeX model using the proposed CatCode framework, providing valuable insights into the current strengths and limitations of these models. Notably, they identify a significant deficiency in the ability of existing models to recognize functionally equivalent code and to preserve information about code functionality when generating explanations or descriptions of the code.

**Questions To Authors:**

- I know it's a naive question but it's an important one. How do you ensure that transformations are applied correctly? Have you tried executing code post AST transformation and obtained the same execution accuracy?

- Given that both Text-Davinci, ChatGPT are OpenAI models and hence are most likely trained on similar sources of data, I am not surprised to see similar performance on multiple tasks. It would be interesting to conduct this analysis with some models other than OpenAI. In particular, can you run experiments with one additional model from Anthropic, Mixtral, Llama models for broader coverage?

**Reasons To Accept:**

- This paper presents a novel and principled approach to evaluating LLMs in the context of code-related tasks, offering a standardized evaluation platform and highlighting important areas for improvement in current models. The proposed approach can be extended to different programming languages and datasets.
- Authors provide granular data about which transformations are hard to be detected by the current SOTA models, guiding community to improve performance in target research areas.

**Reasons To Reject:**

I don't have any major reason to reject this paper but there are a few areas of improvements

- It's important to study if model's ability to handle different transformations improve across the board. However, authors don't present any correlation among performance across transformations. If a model does well on X (variable renaming), does it also do well on (loop exchange)? I think such analysis will add a lot of value to this work.
- Code transformations are extensively studied under robustness areas in code generation space [1] so it's important to connect this paper to robustness literature.




References:
[1] Wang, Shiqi, et al. "ReCode: Robustness Evaluation of Code Generation Models." Proceedings of the 61st Annual Meeting of the Association for Computational Linguistics (Volume 1: Long Papers). 2023.

---

> ### Author Rebuttal · Authors · 2024-05-31
>
> We thank the reviewer for appreciating the novelty and main contributions of our approach, including the "principled method", "standardized evaluation platform", and "granular data on which transformations are hard to detect for LLMs". We address the questions as follows:
> ### Q1: Is there a "correlation among performance across transformations"?
> - This is an insightful question! To think from a correlation aspect is what our framework aims to advocate. For similar morphisms, they're correlated. If a model does well on X (variable renaming), it may not be guaranteed to promote Y (loop exchange) performance because those are two independent morphisms.
> - Here's an interesting relevant fact: if a model does well in "debug then translate", it may likely do well at "translate then debug". (It's nontrivial because they're debugging in different languages. The intuition is that the PL is isomorphic to the Turing Machine under our categorical definition, therefore sharing some structural invariance information). Similarly, why Dall-E can generate many avocado chair images even if they don't exist in the dataset can also be explained by Category Theory.
> ### Q2: Connect this paper to robustness literature.
> We appreciate the advice of including robustness literature as relevant work. We will incorporate references to "ReCode"(Wang et al., 2023) and other literature like "Semantic Robustness of Models of Source Code" (J Henkel et al.,2022), etc. in the next version!
> ### Q3: How to ensure transformation correctness. Does code post-AST transformation obtain the same execution accuracy?
> Yes, we ensure theoretical accuracy based on the definitions of these edits. The correctness of AST transformations can be detected by a tree isomorphism detection algorithm. We also verified them with test cases in the datasets, and obtained the same execution accuracy.
> ### Q4: Conduct this analysis with models other than OpenAI. In particular ... an additional model from Anthropic, Mixtral, Llama models for broader coverage?
> Thanks for the valuable suggestions to improve our work coverage! We've conducted experiments on `Llama2-7b-chat-hf` with key results: (1) Global morphism identification:23% accuracy (2) Translation Java->JavaScript: 59.3% Pass@1 rate on average(HumanEval:38.5, MBXP:43.8, MathQA:95.5) (3) Reproduction: 43.5% Pass@1 on average(HumanEval:25.6, MBXP:45.4, MathQA: 59.4). We will update the results for Mixtral, WizardCoder, StarCoder for a more thorough comparison :)

---

> > ### Comment · Reviewer_hN4m · 2024-06-04
> > **Thanks for additional expriments**
> >
> > Thanks for answering my questions and conducting additional experiments. As indicated by my initial score, I think it's a good paper so I will keep my score with accept decision.

---

> > > ### Author Response · Authors · 2024-06-07
> > >
> > > Dear Reviewer,
> > >
> > > Thank you very much for your supportive feedback and for recognizing the efforts made to address your questions.
> > >
> > > We deeply appreciate your valuable feedback and the opportunity to refine our work further :)

---

### Decision · Program_Chairs · 2024-07-10

**Decision:**

Accept

**Comment:**

This paper presents CatCode, which uses the category theory to design a framework for evaluation of coding tasks. The benefit of this framework is to unify coding benchmarks of different formats, and provide a flexible way to compose new tasks.

The proposed evaluation framework is novel and interesting, and the authors made great efforts to communicate with reviewers during the rebuttal and discussion period. I think this work can be valuable to the conference, and thus recommend acceptance. I encourage the authors to revise the paper and include the results and discussion they promised to add in the author response.